# Structural and Physicochemical Characterization of Extracted Proteins Fractions from Chickpea (*Cicer arietinum* L.) as a Potential Food Ingredient to Replace Ovalbumin in Foams and Emulsions

**DOI:** 10.3390/polym15010110

**Published:** 2022-12-27

**Authors:** Daniela Soto-Madrid, Nicole Pérez, Marlen Gutiérrez-Cutiño, Silvia Matiacevich, Rommy N. Zúñiga

**Affiliations:** 1Food Properties Research Group (INPROAL), Department of Food Science and Technology, Technological Faculty, Universidad de Santiago de Chile, Av. Ecuador 3769, Estación Central, Santiago 9170201, Chile; 2Department of Biotechnology, Universidad Tecnológica Metropolitana, Las Palmeras 3360, Ñuñoa, Santiago 7800003, Chile; 3Molecular Magnetism & Molecular Materials Laboratory (LM4), Department of Chemistry of Materials, Chemistry and Biology Faculty, Universidad de Santiago de Chile, Av. Lib. Bernardo O’Higgins 3363, Estación Central, Santiago 9170022, Chile; 4Center for the Development of Nanoscience and Nanotechnology, CEDENNA, Santiago 9170022, Chile

**Keywords:** plant proteins, chickpea, food ingredients, ovalbumin, total protein extraction

## Abstract

Chickpeas are the third most abundant legume crop worldwide, having a high protein content (14.9–24.6%) with interesting technological properties, thus representing a sustainable alternative to animal proteins. In this study, the surface and structural properties of total (TE) and sequential (ALB, GLO, and GLU) protein fractions isolated from defatted chickpea flour were evaluated and compared with an animal protein, ovalbumin (OVO). Differences in their physicochemical properties were evidenced when comparing TE with ALB, GLO, and GLU fractions. In addition, using a simple and low-cost extraction method it was obtained a high protein yield (82 ± 4%) with a significant content of essential and hydrophobic amino acids. Chickpea proteins presented improved interfacial and surface behavior compared to OVO, where GLO showed the most significant effects, correlated with its secondary structure and associated with its flexibility and higher surface hydrophobicity. Therefore, chickpea proteins have improved surface properties compared to OVO, evidencing their potential use as foam and/or emulsion stabilizers in food formulations for the replacement of animal proteins.

## 1. Introduction

In current times, the consumption of vegetable protein is increasing as consumers seek healthy and lower-cost alternatives to animal protein without compromising product quality, safety, and sustainability. This way, consumer groups, such as flexitarians, vegans, and vegetarians, opt for a diet rich in pulses such as beans, lentils, peas, and chickpeas. In addition, some consumers avoid common plant proteins, such as soy, due to their potential allergenicity and celiac disease or sensitivity [1]. Therefore, the technological development of protein concentrates and isolates derived from legumes, specifically from pulses, is an excellent alternative to meet current consumer demands. These proteins with high functionality could be required concerning stabilizing multiphasic systems, such as foams (e.g., whipped cream and ice cream) and emulsions (e.g., mayonnaise and dressing).

The use of proteins from pulses has grown considerably in the food industry due to their richness in essential amino acids (lysine, leucine, and arginine). Additionally, legumes are an economical protein source and have low water requirements for crops [2]. Therefore, the future food market is focused on food security and the necessity for sustainable protein sources. The Food and Agriculture Organization (FAO) describes legumes as “nutritional seeds for a sustainable future” since crops, such as chickpeas, are resistant to drought and do not require intensive irrigation, becoming a sustainable protein source [3].

Chickpea is the third most abundant pulse crop worldwide, with a high protein content (14.9–24.6%) [4,5]. The literature has reported that chickpea seed proteins are composed of globulin (salt soluble; 56%), albumin (water-soluble; 12%), a prolamin (alcohol soluble; 2.8%), glutelin (acid/alkali-soluble; 18.1%), and residual proteins [4,6]. Compared to other legumes, chickpea proteins have a higher bioavailability [7]; for that, they could be an excellent potential alternative to animal proteins. Chickpea is currently used in the food industry to make bread, sandwiches, soups, pasta, crackers, cakes, beverages, mayonnaise-type dressing, and gluten-free pasta, among other foods [8,9].

The literature has reported three types of vegetable protein extraction techniques: (i) conventional extraction, based on organic and alkaline solvents; (ii) biochemical extraction, based on the use of enzymes and (iii) physical extraction, based on ultrasound, pulsed electric field, microwave or high pressure-assisted extraction [10]. Extraction based on protein solubility using Osborne’s methodology and alkaline extraction is the most used extraction technique in the food industry to obtain protein concentrates and isolates [11,12,13]. It is due to the low cost of chemical products, the relative simplicity of the critical apparatus, and the use of environmentally friendly solvents.

Osborne’s methodology sequentially extracts proteins based on their solubility and subsequent precipitation at the isoelectric point. Albumins are soluble in water; globulins are insoluble in water but soluble in dilute salt solutions; glutelins are insoluble in the above solutions but soluble in weak acid or basic solutions; and finally, prolamins are insoluble in the above solutions but soluble in alcohol/water mixtures [14]. This methodology was successfully used for the sequential extraction of proteins from rice, quinoa, and chickpeas [4,12,15].

Using isolated and/or protein fractions from pulses in food matrices depends on their composition and functional and structural properties [16,17]. However, the type and variety of legume seeds and the extraction methods can alter the protein composition in the final isolated or concentrated products [1,18]. Most studies on pulses protein isolates’ structural and functional properties have yet to consider the role of the individual protein fractions (e.g., albumin, globulin, glutelin) compared to the total protein extracted. Chang et al. [17] evaluated the structural and functional properties of protein fractions such as globulin, legumin, and vicilin from green peas and chickpeas, observing differences in the functionality of the protein fractions. However, the authors did not consider the behavior of the total protein fraction. For that, evaluating the functional and structural properties of the total and each protein fraction is essential to provide knowledge to manufacture suitable pulse proteins according to the application area. Thus, it becomes a potential ingredient for the total or partial replacement of animal proteins.

In this study, chickpea proteins were compared to ovalbumin, an animal protein widely used as an emulsifying agent in the food industry, because of its intense surface activity and emulsion stabilization properties [19]. In addition, due to its high foaming property, it is used as a foaming agent to improve and maintain the quality (texture and volume) of aerated foods, such as cakes, cookies, dessert shells, and chocolate mousses [20]. However, it has disadvantages compared to vegetable protein due to the high cost, limited supply, and direct relationship to climate change [10]. For that, it is crucial to obtain alternative vegetable sources of proteins with functional properties similar to animal-origin proteins to replace them and develop appropriate technologies for their profitable extraction and incorporation as an ingredient in foods.

Today, the food industry calls for research and development in plant-derived protein as a versatile and inexpensive substitute for animal protein in the human diet. Therefore, this study aims to evaluate the surface and structural properties of protein fractions (by total and sequential extraction) isolated from defatted chickpea flour, compared to an animal protein, ovalbumin, for its potential use as a food ingredient. For this purpose, extraction, fractionation, and physicochemical characterization were performed for each protein fraction to evaluate the quality of fraction proteins based on their amino acid profile and potential food application.

## 2. Materials and Methods

### 2.1. Samples

Chickpea flour (Extrumol, Santiago, Chile) was characterized by proximal composition (g/100 g): 8.4 ± 0.1 of moisture, 18.3 ± 0.3 of protein, 6.9 ± 0.1 of lipids, 2.9 ± 0.0 ash, 0.9 ± 0.0 crude fiber, and 62.6 ± 0.3 non-nitrogen extracts. This raw material was defatted before the protein extraction process.

Commercial ovalbumin (OVO) with a protein value of 38.2 ± 0.1 g/100 g was purchased from Cherry (Chile) for comparison as a control surface and structural properties. 

Defatted chickpea flour: Hexane of analytical grade (Heyn, Santiago, Chile) was used to eliminate the lipid fraction from the chickpea flour by solvent extraction at a ratio of 1:3 weight: volume (flour:solvent) for 1 h under stirring at 200 rpm. Subsequently, the mixture was filtered and dried under an extractor hood at room temperature for 24 h. The defatting process of the chickpea flour was repeated twice, according to the methodology described by Karaca et al. [21]

### 2.2. Protein Extraction Processes from Defatted Chickpea Flour

#### 2.2.1. Total Extraction (TE)

The total protein fraction (Figure 1) was obtained by isoelectric precipitation following the protocol of Chang et al. [22] with some modifications. For this, 20 g of defatted chickpea flour was mixed with 200 mL of NaOH 0.02% *w/v* (Heyn, Chile) at a pH of 11.5 and 400 μL of protease inhibitor (Sigma-Aldrich, Darmstadt, Germany), according to the methodology of Castellión et al. [12]. The mixture was stirred for 1 h with continuous stirring at a speed of 200 rpm (Boeco, model MSH 420, Hamburg, Germany). Then, the mixture was centrifuged at 4130× *g* for 10 min (Restek, model sep-3000, PA, USA), and a recycling process was carried out, where the sample was remixed with 200 mL of NaOH at 0.02% *w/v* for 1 h. The supernatant was filtered with the help of a vacuum pump (Rocker, model 300C, Kaohsiung, Taiwan). Subsequently, it adjusted the pH of the supernatant to 4.5 with 2 N HCl (isoelectric point of legumin proteins). The supernatant was centrifuged at 4130× *g* for 10 min, a process that was repeated twice, washing the pellet with purified water, and then resuspending it in purified water. The total protein obtained was stored at −80 °C in an ultra-freezer (Thermo Scientific, model 702, MA, USA) and then freeze-dried at −40 °C and 27 Pa (Virtis SP Scientific, Benchtop Pro 9L ES-55, PA, USA). After the freeze-drying process, the moisture and water activity of the TE sample were determined. Finally, the freeze-dried proteins were stored in desiccators until further analysis.

#### 2.2.2. Sequential Extraction (SE) of Proteins from Chickpea Flour

Albumin (ALB), globulin (GLO), and glutelin (GLU) fractions were extracted sequentially according to the procedure of Chang et al. [4] with some modifications (Figure 2). A sample of defatted chickpea flour (50 g) was mixed with purified water (200 mL) and 1 mL of protease inhibitor (Sigma-Aldrich, Darmstadt, Germany), according to the methodology of Castellión et al. [12]. The mixture was magnetic stirred (Boeco, model MSH 420, Hamburg, Germany) for 2 h and then centrifuged at 4130× *g* for 10 min (Restek, model Q-SEP 3000, PA, USA). The supernatant was filtered using Whatman N°1 paper and a vacuum pump (Rocker, model 300C, Kaohsiung, Taiwan), while the precipitate was reserved for GLO extraction (pellet 1). The filtered supernatant was precipitated at pH 4.1 with 1 M HCl, centrifuged at 4130× *g* for 10 min (Restek, model Q-SEP 3000, PA, USA), and subsequently freeze-dried (Virtis SP Scientific, Benchtop Pro 9L ES-55, PA, USA) obtaining the ALB fraction. Pellet 1 was mixed with 200 mL of a NaCl solution (5% *w/v*) and magnetically stirred for 2 h. It was centrifuged at 4130× *g* for 10 min, the supernatant was filtered in the same way as carried out previously, and the precipitate was reserved (pellet 2). The supernatant was precipitated at pH 4.3 with 1 M HCl, centrifuged again at 4130× *g* for 10 min, and the precipitate corresponding to the GLO fraction was also freeze-dried. Pellet 2 was mixed with 200 mL of a NaOH solution 0.1 M, stirred for 2 h, and centrifuged at 4130× *g* for 10 min. The supernatant was filtered and precipitated at pH 4.8 with 1 M HCl, centrifuged at 4130× *g* for 10 min, and freeze-dried, obtaining the GLU fraction. Finally, all the freeze-dried proteins (ALB, GLO, and GLU) were stored in desiccators until further analysis. 

#### 2.2.3. Moisture Content and Water Activity of Freeze-Dried Proteins

The moisture content of the freeze-dried proteins was determined gravimetrically by the difference in mass before and after drying the samples in an oven (Memmert, model WNB 22, Schwabach, Germany) at 105 °C until constant weight (AOAC, 1998). Results were expressed as dry basis percentage (% d.b; g water/100 g solids). The water activity (a_w_) was determined by automatic measuring equipment (Novasina, model Lab Start, Lachen, Switzerland) at 25 °C [23].

#### 2.2.4. Protein Extraction Yield

The yield (Y) of the total and sequential extractions of chickpea proteins was estimated by a gravimetric method based on the initial protein mass and the mass of freeze-dried protein obtained after the extraction process, according to the following equation.
(1)$$Y\;\left(\%\right)=\frac{\mathrm{mass}\;\mathrm{of}\;\mathrm{freeze}-\mathrm{dried}\;\mathrm{protein}\;\left(g\right)}{\mathrm{mass}\;\mathrm{of}\;\mathrm{protein}\;\mathrm{in}\;\mathrm{chickpea}\;\mathrm{flour}\;\left(g\right)}\times 100\%$$
where the protein mass of the flour on a dry basis was 16.8 g/100 g of sample (corrected for moisture content), according to its proximal analysis. 

### 2.3. Physicochemical Characterization of the Protein Fractions 

#### 2.3.1. Sodium Dodecyl Sulfate-Polyacrylamide Gel Electrophoresis (SDS-PAGE)

Electrophoresis was carried out according to the methodology described by Laemmli [24]. The molecular weight of the proteins was estimated using SDS-PAGE polyacrylamide gels (Bio-Rad, Mini-protein TGX 12% gel, CA, USA) compared to standard proteins (Precision Plus Protein Kaleidoscope Standards, Broad Range, Bio-Rad Laboratories Inc., CA, USA). Protein samples (0.3% w/w) were prepared by dissolving 50 µL of protein dispersion in 50 µL of a loading buffer (50 μL of β-mercaptoethanol (BME) per 950 μL 2X Laemmli sample buffer) (Bio-Rad Laboratories, Inc., CA, USA). The samples prepared in the previous step were heated at 95 °C for 5 min in a water bath for protein denaturation. Samples (20 µL) were loaded into each well. The electrophoretic migration was performed using refrigerated running buffer (1 L), which was prepared using 1X Tris-acetate-EDTA (TAE) buffer from 10X Tris R11 glycine/SDS buffer (Bio-Rad Laboratories Inc., CA, USA). This process was monitored at a constant current (100 V) for 1.5 h using a Mini-Protean Tetra Cell unit (Mini-PROTEAN^®^ Tetra System, Bio-Rad Laboratories Inc., Richmond, CA, USA). The gel was stained with a Coomassie blue solution (Bio-Rad, Coomassie Brilliant Blue R-250, Bio-Rad Laboratories, Inc., CA, USA) for 12 h at room temperature. After the staining time, the solution was removed, and 200 mL of the destain solution I (10% acetic acid *v/v*, 50% methanol *v/v*) was added while gently stirring for 2 h. Then, it was removed, and 200 mL of the destain solution II (7% *v/v* acetic acid, 5% *v/v* methanol) was added and stirred for 12 h at room temperature. Finally, gels are stored in distilled water until digital images were taken. All reactive stains were of analytical grade (Heyn, Santiago, Chile).

#### 2.3.2. Amino Acid Analysis by High-Performance Liquid Chromatography (HPLC)

The chickpea protein’s amino acids were identified and quantified according to the methodology described by Janssen et al. [25]. Chickpea proteins (10 mg) were hydrolyzed with 300 µL of a 6 N HCl solution at 110 °C for 24 h. The hydrolyzate obtained was derivatized with 20 µL of phenylthiocyanate (10% *w/v*) to generate phenylthiocarbamyl amino acids, which were separated and quantified by HPLC at 254 nm. A liquid chromatograph (Waters 600 controller, MA, USA) with a diode array detector (Waters 996) and a Phenomenex (Los Angeles, CA, USA) RP18 column (150 mm × 4.6 mm, 5 µm) was used. The gradient separation was performed using two solvent solutions: Solution (A) composed by 94:6 *v/v* of 0.14 mol/L of HPLC-grade anhydrous sodium acetate (pH 5.9): HPLC-grade acetonitrile, and solution (B) by HPLC-grade acetonitrile: water (60:40 *v/v*) solution. The injection volume was 20 µL, the column temperature was 40 °C, and the analysis time was 30 min. The quantification of amino acids was carried out with external standards (Sigma-Aldrich, Darmstadt, Germany) and attributed to 17 amino acids previously described in chickpeas [26].

#### 2.3.3. Zeta Potential of Protein Dispersions

The pH stability and isoelectric point determination were carried out through Zeta potential measurements (Zetasizer Nano Series, Nano ZS90, Malvern Instruments, UK). TE, ALB, GLO, and GLU dispersions (0.1% *w/v*) were prepared at the respective solubility solvent; with ultrapure water, 5% (*w/v*) 0.1 M NaCl and NaOH, respectively. pH values were adjusted in the range of 2 to 7, using NaOH or HCl (1 N). Samples of protein dispersions (1 mL) were placed in a cuvette with electrodes (Malvern Instruments, cell type DTS1070, UK), and Zeta potential was measured. Zeta potential versus pH was plotted to obtain the isoelectric point (IP), which is the point where the value of the net surface charge of proteins (and Zeta potential) is equal to zero, and +/−30 mV values are required for complete electrostatic stabilization [27]. 

### 2.4. Surface and Structural Properties of Protein Dispersion

#### 2.4.1. Surface Hydrophobicity

Protein fractions were measured using 1-anilino naphthalene-8-sulfonic acid (ANS) as a fluorescence probe, as described by Kato & Nakai [28]. Samples were diluted between 0.002 and 0.01% (*w/v*) from a concentrated dispersion of proteins (TE, ALB, GLO, and GLU) at 0.1 (*w/v*). Four milliliters of diluted samples reacted with 20 µL of ANS solution (8 × 10^−3^ M). The fluorescence intensity was measured at the excitation wavelength of 365 nm and the emission wavelength of 484 nm on a fluorometer (Perkin Elmer, LS55, Massachusetts, UK). Fluorescence intensity and the corresponding concentration of protein were fitted using linear regression. The slope of the curve obtained was defined as the surface hydrophobicity index of the protein.

#### 2.4.2. Dynamic Interfacial (DIT) and Surface Tension (DST) Measurements

Interfacial or surface tension changes between protein dispersions (TE, ALB, GLO, and GLU) were determined by an optical tensiometer (Ramé-Hart Inc., model 250-F4, Succasunna, NJ, USA). Commercial ovalbumin (OVO) was used as a control. All proteins were analyzed at a concentration of 0.1% (*w/v*), previously hydrated at 4 °C overnight, and, subsequently, the pH value was adjusted at 7 before measurements. Measurements were based on the pendant drop method using a lipid phase (sunflower oil) for DIT and air for DST. In this method, an axisymmetric drop of protein dispersion (8.5–10 μL) was delivered and allowed to stand at the tip of a steel needle inside a quartz cell with 30 mL of sunflower oil or air at 25 °C to achieve protein adsorption at the oil-water or air-water interface, respectively. For DIT measurements, commercial sunflower oil (Natura, Argentina) was purified by resin mixing (Florisil^®^ 60-100 mesh, 46385, Sigma-Aldrich, Germany) at a 10:1 ratio for 4 h, according to the methodology of Bahtz et al. [29].

The Coupled Charged Device (CCD) camera of the optical tensiometer captured drop images at different time intervals. The interfacial or surface tension was calculated by analyzing the image profile of the drops stabilized by the surfactant dispersions using image analysis (Ramé-Hart Inc., DROPimage Advanced software, Succasunna, NJ, USA) and then by fitting the Laplace equation to the drop shape. To validate the DIT methodology, the interfacial tension of the sunflower oil/pure water system was the same as previously reported (26.6 ± 0.5 mN/m) for identical conditions [30]. For DST measurements, it corroborated that the surface tension of the pure water/air system was 72.8 ± 0.2 mN/m [31].

Results obtained from the DIT and DST measurements were interpreted in terms of interfacial pressure or surface pressure, respectively, which was defined as the decrease in interfacial or surface tension of a pure solvent caused by the addition of the protein:(2)$$\Pi =\tau -\tau_{p}$$
where Π is the interfacial or surface pressure of the protein dispersion (mN/m), τ is the interfacial tension of the oil-pure water system (26.6 mN/m), or the surface tension of the air-pure water system (72.8 mN/m) at 25 °C, and τ_p_ is the interfacial or surface tension of the protein dispersion (mN/m) at the same temperature.

The evolution of interfacial or surface pressure was fitted to an empirical kinetic model based on the works of Azuara et al. [32] for the osmotic dehydration process and Moyano & Pedreschi [33] for oil absorption during deep-fat frying. A balance in terms of the interfacial or surface pressure at the interface is performed as follows.
(3)$$\Pi =\Pi_{0}+\Pi_{\mathrm{eq}}-\Pi^{*}$$
where Π_0_ is the interfacial or surface pressure at time zero (mN/m), Π_eq_ is the interfacial or surface pressure at the equilibrium (mN/m), and Π is the interfacial or surface pressure value needed to reach the equilibrium (mN/m). The limit conditions for interfacial or surface pressure are:

(i) At t = 0 → Π* = Π_eq_ and Π = Π_0_(ii)At t = ∞ → Π* = Π_0_ and Π = Π_eq_

It is assumed that changes in the ratio Π^*^/Π are inversely proportional to protein adsorption time:(4)$$\frac{\Pi^{*}}{\Pi }=\frac{1}{k\times t}$$

Considering the above conditions, the empirical model proposed is:(5)$$\Pi =\frac{\Pi_{\mathrm{eq}}\times k\times t}{1+k\times t}+\Pi_{0}$$
where k is the specific rate constant for protein adsorption to the interface (s^’1^). Interfacial or surface pressure curves were fitted to the proposed model. The specific rate constant (k) and the interfacial or surface pressure at equilibrium (Π_eq_) were obtained by employing nonlinear regression analysis performed with the Solver tool of Microsoft Excel, using as an objective function the minimization of the root mean square (RMS) equation.
(6)$$\mathrm{RMS}\;\left(\%\right)=\sqrt{\frac{1}{n}\times \sum {\left(\frac{V_{e}-V_{p}}{V_{p}}\right)}^{2}}\times 100\%$$
where n is the number of data points for each curve, V_e_ is the experimental value, and V_p_ is the predicted value by Equation (6).

#### 2.4.3. Structural Characterization by Fourier Transform Infrared-Attenuated Total Reflectance (FTIR-ATR) Spectra Analysis

Chemical groups and bonding arrangement of components present in protein samples were determined by Fourier Transform Infrared-Attenuated Total Reflectance (FTIR -ATR), using a Jasco FTIR-4600 spectrophotometer equipped with an ATR PRO ONE (Jasco, Easton, MD, USA). Measurements were performed in a spectral range of 4000 to 400 cm^−1^ at a 4 cm^−1^ resolution and scan number 32. Fourier second derivative analysis was performed for the Amide I region (1700–1600 cm^−1^) using the OriginPro 8.5 software (OriginLab, Northampton, MA, USA). Curve normalization was developed at the highest intensity peak and Gaussian peak fitting using OriginPro 8.5 software (OriginLab, Northampton, MA, USA). The percentages of the secondary structures were determined by integrating the areas of the fitted peaks. Intensity measurements were performed on the original and the second-derived spectra by calculating the height of the absorbance bands from their baseline. All chemical functional groups were identified using published reports [9,17,34].

### 2.5. Statistical Analysis

All experiments were run in triplicate. Data were reported as means with their corresponding standard deviation. ANOVA test, at a confidence level of 95%, was performed to determine statistical differences using Statgraphics Centurion XVI^®^ software (StatPoint Technologies Inc., VA, USA). Differences between samples were evaluated using multiple range tests, using the Least Significant Differences (LSD) multiple comparison method. The significance of the differences was determined at a 95% confidence level (*p* < 0.05).

## 3. Results and Discussion

### 3.1. Protein Extraction Processes

#### 3.1.1. Effect of Process Parameters in Protein Fractions

Since lipids can interfere as a barrier to solvent penetration during protein extraction [11], a defatting procedure was previously applied to the flour. The initial lipid content of the flour was 6.9 g/100 g of wet basis sample (w.b.). The final lipid content of the defatted flour was 0.82 ± 0.01 g/100 g of sample (w.b.). Therefore, the defatted process was successful, and in this case, the lipid content was a process parameter not considered in the extraction process yield.

Moisture content and water activity (a_w_) are relevant physical parameters to evaluate the freeze-drying process of protein extracts since moisture content and a_w_ influence their stability and safety during storage. Table 1 shows the moisture content of the extracted chickpea protein samples. All values were less than 4% w.b., which ensures stability while storing freeze-dried proteins, preventing particle agglomeration and caking [35]. All freeze-dried extracted protein samples (TE and SE) presented a_w_ values lower than 0.22 (Table 1), ensuring their preservation since most microorganisms grow at a_w_ values higher than 0.6 [36]. Therefore, the freeze-drying process of the TE, ALB, GLO, and GLU samples was optimal. Obtaining powdered protein extracts could allow their potential use as an ingredient in food applications, suggesting stable storage.

#### 3.1.2. Protein Extraction Yield

Table 2 indicates no significant differences (*p* ≥ 0.05) in the process yield for both extractions (TE and SE), which was 82 ± 3% (value obtained through the average of both extractions). These results were superior to those reported by [37], ranging between 50.3–69.1%, which depended on the chickpea variety (Kabuli and Desi) or protein extraction methodology. Sánchez-Vioque et al. [11] reported that protein recovery ranged from 62.1% to 65.9% for two protein isolates from ground Kabuli chickpea seeds. The differences in protein extraction yield values between authors can be attributed to raw materials, protein solubility, and processing times. However, the high value obtained in this study confirms this suitable methodology for TE.

In the SE (Table 2), ALB (64%) was the protein extracted with the highest proportion, followed by GLO (8%) and GLU (6.5%), obtaining a total protein extraction of 85 ± 6%. However, previous literature indicated that chickpea protein mainly comprises GLO (about 50%) [22,38,39]. In contrast, the highest percentage of ALB obtained (Table 2) could depend on the extraction conditions and physicochemical procedures used [40]. In this sense, Liu et al. [41] carried out ALB and GLO extractions from different chickpea varieties using solutions of water and salt (K_2_SO_4_ and NaCl). They obtained high concentrations of ALB (~60%) when using NaCl as an extraction solvent, similar to this study, because legumes are a source of minerals, especially a richer source of calcium and phosphorus than most cereals [2]. Therefore, the salts in the chickpea flour could convert the purified water used as the extraction medium into a very dilute saline solution, extracting significantly higher ALB and GLO content with distilled water.

### 3.2. Physicochemical Characterization of Total and Sequential Protein Extracted from Chickpea Flour

#### 3.2.1. Electrophoretic Pattern of Chickpea Protein Fraction Evaluated by SDS-PAGE

SDS-PAGE was used to determine the molecular weight (Mw) of the subunits of the chickpea protein fractions. After electrophoresis of TE, ALB, GLO, and GLU proteins, multiple bands were observed and compared to a Mw standard (Figure 3 lane 1), ranging from approximately 97 to 12 kDa (Figure 3). These values agree with the characteristic band patterns of legumes reported by several authors [4,6,22], which indicates the high effectiveness of the protein extraction process from the chickpea flour.

In Figure 3, the SDS-PAGE patterns of the TE (lane 2) and of the ALB fraction (lane 3) were very similar, observing the expected proteins for both fractions: legumin (11S), vicilin (7S) and albumin (2S) [4]. Legumin protein (α and β subunits) were slightly observed at ~46 kDa and ~25 kDa bands, respectively [4,11]. Furthermore, estimated molecular weights of ~49, ~35, ~33, ~19, and ~15 kD were identified as chickpea vicilin (7S) subunits reported by Chang et al. (2011). Finally, low molecular weight bands ~12 kD were attributed to albumin subunits (2S) [42]. These results indicate that the TE sample contains a high percentage of proteins extracted on the ALB fraction due to the similarity in the band pattern, which agrees with the extraction yield results (Table 2). 

In the GLO fraction (Figure 3, lane 4), a ~98 kDa molecular weight (MW) band was identified, compared to the chickpea lipoxygenase, as Clemente et al. [43] reported. However, a similar lighter band pattern was observed for GLO (lane 3) and GLU (lane 4) fractions, attributed to the chickpea 11S and 7S subunits.

Therefore, as expected, the pattern of bands for the TE corresponds to the mixture of the extracted proteins. The ALB, GLO, and GLU fractions present a similar pattern of bands but with low resolution compared to other legumin proteins [17]. 

#### 3.2.2. Amino Acid Profile of Extracted Chickpea Proteins

Proteins are chains of amino acids linked by peptide bonds; these amino acids provide nutritional value and influence their structure and functionality. Table 3 shows the 17 amino acids detected by HPLC-DAD for each extraction sample from chickpea flour. The highest amino acids percentages in the TE, ALB, GLO, and GLU samples were glutamic acid, aspartic acid, and arginine, where the glutamic acid presented the highest proportion in all samples (TE: 14.5 ± 0.3; ALB: 11.5 ± 1.3; GLO: 8.3 ± 0.6, GLU: 9.7 ± 0.1 g/100 g protein). These results agree with Ghribi et al. [44] since, in chickpea protein isolates, glutamic acid showed the highest amount, varying from 15.04 to 19.23 g/100 g of protein. Furthermore, the total essential amino acids content was similar to that reported by Chang et al. [17] (~25 g/100 g protein).

The percentage of essential amino acids (EEA%) of the chickpea proteins detected was calculated (Sum of detected essential amino acids/total amino acids content × 100%). It did not determine the content of Tryptophan because it was hydrolyzed during the acid hydrolysis step of the methodology used. No significant differences were obtained between GLO and GLU samples (36.0 ± 0.1%) and between TE and ALB (33.5 ± 0.9%). It is essential to remark that the total essential amino acid determined for isolated chickpea after total and sequential fractions could reach the FAO/WHO [45] requirement for the essential amino acid for preschool children. Consequently, all protein fractions obtained in this study could be potentially applied to food matrices as a good source of quality proteins. 

The amino acid profile of the extracted proteins shows significant differences (*p* ≤ 0.05) related to hydrophobic amino acids quantity (TE: 25.4 ± 0.2; ALB: 24.33 ± 1.27; GLO: 19.71 ± 0.03; GLU: 23.24 ± 0.42 g/100 g protein). The TE, ALB, and GLU samples presented the highest values, which shows their potential use as foam and/or emulsion stabilizers. The literature has indicated that globular proteins stabilize air/water interfaces in foams and oil/water interfaces in emulsions, where hydrophobic amino acids play a fundamental role [46,47].

Thus, the amino acid composition can affect the functional properties of proteins because it determines the ratio of hydrophilic and hydrophobic groups and the balance of positive, negative, and neutral groups, which affects their hydrophobicity at the surface and electrostatic interactions. The presence of charged patches along proteins and/or polyelectrolytes in the isolated fractions could also affect their stability [48]. These parameters must be considered when characterizing the protein’s functional performance.

#### 3.2.3. Zeta Potential of Extracted Chickpea Proteins

The measurement of the Zeta potential gives information on the dispersion stability. The highest Zeta potential of suspensions (more positive than +30 mV or negative than -30 mV) showed physical stability. It is due to charged particles repelling each other and overcoming the natural tendency to aggregate [27,49]. Figure 4 shows the Zeta potential of the protein fractions extracted from chickpea flour. The TE and ALB samples at pH 7 and 3 present the highest absolute values, respectively (TE: |36.4| ± 0.6 mV; |30.4| ± 1.4 mV and ALB: |30.0| ± 1.8 mV; |29.5| ± 1.4 mV), evidencing the stability of protein dispersions at this pH and may have a potential application in food matrices at neutral and acidic conditions.

On the other hand, the GLO fraction presented the lowest Zeta potential range (+7 and –7 mV) for the range of pH studied, followed by the GLU fraction, where the literature has established that potentials between |5| and |15| are in the limiting flocculation region [27]. This low stability for GLO and GLU fraction was expected, considering that extraction solvents contained polyelectrolytes (NaCl and NaOH, respectively). Their presence could affect protein stability attributed to the protein-polyelectrolytes complex mechanism based on charge patch and charge regulation [48]. 

In parallel, Zeta potential results show an isoelectric point (IP) close to 4.5 for all proteins (Figure 4), which confirms the choice of a pH = 4.5 for protein precipitation according to the method of extraction employed. This result also agrees with Vani & Zayas [50], who indicate that most vegetable proteins have an IP between 4.0–5.0. It is essential to consider that proteins could adsorb charges on their surfaces at their IP, even when their net charge is neutral. Second or higher-order electrostatic factors could also favor their adsorption/interaction and stability [48]. Based on the above, TE and ALB would potentially apply in food formulations with pHs higher or lower than the IP.

### 3.3. Surface and Structural Properties of Vegetable Protein Dispersion Compared to Animal Protein: Ovalbumin

#### 3.3.1. Surface Hydrophobicity

Protein surface hydrophobicity characterizes the number of hydrophobic groups on the surface of protein molecules in contact with the polar water environment. It can affect many functional properties, such as emulsification, foaming, and gelation. It is an important index for the physical, chemical, and functional properties [51], which can also measure the degree of denaturation of proteins. This work compared the surface hydrophobicity of the extracted chickpea protein fractions with the food industry’s most widely used animal protein: ovalbumin. This protein showed similar characteristics to chickpea proteins, such as a globular protein, high solubility in aqueous media, and isoelectric point at pH 4.6 [52]. The surface hydrophobicity was measured at pH = 7, considering potential applications in foods, due to the high stability in all fractions observed at neutral and acidic pH (Figure 4).

All the extracted proteins showed a higher surface hydrophobicity index than ovalbumin (OVO) (Figure 5). The index values for TE, ALB, GLO, and GLU were 3.7, 4.0, 5.7, and 2.2 times higher than OVO. The higher surface hydrophobicity of plant proteins was not expected because all fractions contain 11S subunits. Hexamers are hydrophobic and located within the macromolecular assembly of the legumin, while the less hydrophobic acid domains are located on the surface [53]. In this case, the balance of forces to maintain the legumin structural domains is disturbed, for example, by the dehydration step of the isolate preparation process, which modifies the final surface hydrophobicity of the protein. This was demonstrated, since the protein presented some degree of denaturation during the extraction process, which is corroborated by the SDS-PAGE (Figure 4, lane 4), where the 11S subunit showed a light pattern. 

Figure 5 also indicated that GLO samples showed the highest surface hydrophobicity index (30,110 ± 2248 a.u.) compared to other fractions. It is important to note that surface hydrophobicity is a structure-related function. The above depends on the size and shape of the protein molecule, amino acid composition, sequence, and intra- e intermolecular cross-link [54,55]. So, in this case, it cannot be attributed to a high hydrophobic amino acid concentration (Table 3) and sequence, considering that no significant differences were observed between protein fractions. Although the GLO fraction is mainly composed of 11S protein, it is not enough to increase this parameter significantly. For that, the relationship between spatial conformations and surface hydrophobicity could be determined by FTIR as a shift in the wavelength of amide II, which is related to secondary structural conformation [56,57].

#### 3.3.2. Dynamic Interfacial (DIT) and Surface Tension (DST)

Amphiphilic molecules, such as proteins, are recognized for their ability to reduce interfacial and surface tension between the dispersed and continuous phases, which is essential for stabilizing emulsions and foams. In recent decades, the literature has reported the wide use of partially hydrophobic molecules for the stabilization of foams and emulsions due to their ability to adsorb at air-liquid strongly and liquid-liquid interfaces to sterically hinder coarsening [58,59,60].

In this sense, the interfacial behavior of chickpea proteins (TE, ALB, GLO, and GLU) was evaluated in comparison with the animal protein OVO. The pendant drop method measured dynamic interfacial (DIT) and surface tension (DST). The results were expressed as interfacial and surface pressure and fitted to a kinetic model with excellent performance, with all RMS values ≤ 5.6% (Table 4). From this model, the specific rate constant for protein adsorption at the interface (k) and the equilibrium pressure (Π_eq_) were obtained (Table 4). 

The rate at which proteins are absorbed in an interface is essential during emulsions and foams’ formation. Concerning the interfacial pressure (Table 4), all the chickpea proteins extracted in this work presented a lower absorption rate (k) than OVO. When analyzing Π_eq_, GLO showed the highest value (19.70 ± 0.27 mN/m) (Figure 6b), indicating that this protein may have potential use in stabilizing emulsions. For surface pressure (Table 4), TE, ALB, GLO, and GLU fractions presented lower absorption rate values at the interface than OVO. The protein that showed the highest Π_eq_ value (34.29 ± 0.42 mN/m) (Table 4) was GLU fraction, which could positively affect foam stabilization. It can explain because chickpea proteins showed a rapid diffusion and absorption at the oil/water and air/water interface compared to OVO, followed by a slower stage in which the conformational rearrangement of the proteins occurs at the interface.

Molecular weight has been reported to influence the adsorption kinetics of amphiphilic molecules at the oil/water and air/water interface. Low molecular weight surfactants can diffuse faster and lower interfacial tension to a greater extent than high molecular weight surfactants. However, high molecular weight surfactants are more effective in forming a viscoelastic film surrounding oil droplets or gas bubbles, which favors the stabilization of emulsions and foams [61,62]. Those results agree with the results obtained in this study since the Mw profile of the OVO was in the range of 103 kD and 38 kDa (data not shown), values higher than the band profile for all chickpea proteins, which were concentrated between 49 and 12 kDa (Figure 3). Hence, TE, ALB, GLO, and GLU had a lower Mw than OVO. Therefore, chickpea proteins have improved surface and interfacial behavior compared to OVO, evidencing their potential as foam and emulsion stabilizers for proteins extracted (GLO and GLU) and may be a replacement alternative to animal protein.

#### 3.3.3. Structural Properties: Secondary Structure Analysis

FTIR spectra of OVO, TE, ALB, GLO, and GLU proteins are shown in Figure 7a. A similar pattern was observed between samples, where the peak at 1050 cm^−1^ showed the OVO characteristic band corresponding to the sulfoxide (S=O) bond [63], having a significant intensity compared to plant proteins. All samples had a broad and robust peak between 3300–3500 cm^−1^ called amide A, representing the intermolecular H-bonded, O-H, and N-H stretching vibrations [64]. As expected in all the samples, FTIR showed two intense bands around ~1633 cm^−1^ and ~1520 cm^−1^ corresponding to the amide I (C=O stretching) and amide II (NH bending and CN stretching) regions of the proteins, respectively [6,65]. In addition, it also identified a band around ~1233 cm^−1^ corresponding to the amide III region (CN stretching, NH bending) [65]. 

The amide I absorption region in the infrared spectrum of a protein is helpful for secondary structure elucidation since the amide I band is the sum of overlapping component bands (α-helix, β-sheet, β-turn, and randomly coiled conformation). It is mainly related to the C=O stretching of the peptide bonds influenced by their various environments in the different kinds of secondary structures [65,66]. Therefore, knowing the secondary structural composition of the proteins analyzed (OVO, TE, ALB, GLO, and GLU) is relevant to understanding their behavior for technological applications. Previous studies have shown that the α–helix and β–sheet protein secondary structures are responsible for emulsion and foaming capacities [67].

It is important to note that a second derivative analysis is often performed before deconvolution to identify the peaks required for peak fitting. The current study’s second derivative peaks were attributed to the secondary structure peak assignments (Figure 7b), following those reported by Chang et al. [17]. Normalization of the amide I region was also performed to obtain a good peak shape for peak fitting and comparison between samples, considering the highest intensity equal to 1. Later, it was deconvoluted using “Gaussian peak fitting”. Each sample revealed 10 major Gaussian bands (Figure 7b) whose wavenumbers corresponded to β-sheet (~1620, ~1625, and ~1630 cm^−1^), random coils (~1641 cm^−1^), α-helix (~1651 and ~1659 cm^−1^), β-turn (~1668 and 1675 cm^−1^), anti-parallel β-sheet (~1681 cm^−1^), and aggregates (~1692 cm^−1^) [9,17,34]. For all samples, amide I peak deconvolution showed a secondary structure composition of β-sheet (~31%), followed by α-helix (~20%), β-turn (~18%), and anti-parallel β-sheet (~10%). These results were consistent with previous studies of chickpea protein [68]. Therefore, it is correct to compare animal and plant proteins used in this study due to their similar composition of structural conformation.

However, changes in peak area (1700–1600 cm^−1^) were observed, which influenced the secondary structure of proteins. Generally, the α-helix and β-sheets are buried inside the polypeptide chains and utilize intermolecular hydrogen bonds between carbonyl oxygen (-CO) and amino hydrogen (-NH) to stabilize the secondary structures [67].

Meanwhile, β-turn and random coils originate from unfolding highly ordered protein structures and impart flexibility to proteins [69,70]. The ratio β-turn/random coil was higher in legumin protein fractions (1.8–2.0) compared to OVO (1.5), demonstrating that the extracted protein fractions could be more flexible than this animal protein. This behavior was previously report using the ability of ultrasound to convert random coil and β-turn into stable and orderly helical structures stabilized by hydrogen bonds but converted at different rates [71,72]. It is important to remark that cavitation-induced protein unfolding, and dissociation can coincide with aggregate formation [73]. The Amida I/Amida II ratio [67] indicated a shifted wavelength for all fractions compared to OVO (1.077), the highest shift for GLO of 1.067. Consequently, the most significant conformational changes in secondary structure protein were observed in the GLO fraction due to Amida II shift wavelength from 1516 (OVO) to 1530 cm^−1^ (GLO). This result confirms the highest surface hydrophobicity of this fraction.

On the other hand, when comparing the intensity results of secondary structure for α-helix, the samples TE, GLO, and GLU (20.3, 20.5, and 20.0%) present lower values than OVO (21.1%). It could be related to differences in the level of flexibility and functional properties [74,75]. Besides this, differences were observed in the intensity of the peak attributed to the β-sheet comparing OVO and TE. The intensity ratio between peaks at 1633 and 1624 cm^−1^ were 1.7 and 1.36–1.44 for OVO and TE and fractions, respectively, indicating significant differences between the conformational structure of the β-sheet of animal and plant proteins. Studies by Zhu et al. [74] and Yan et al. [75] indicated that a low content of α-helix provides high flexibility and good emulsifying and foaming properties in vegetable proteins such as soybean, where flexible proteins showed more robust surface activity than rigid ones. The above was related to the good results for interfacial pressure, surface pressure, and surface hydrophobicity that chickpea proteins presented compared to animal proteins. They significantly decreased the absorption rate at the oil-water and air-water interface (Table 4) and showed higher values of surface hydrophobicity (Figure 5) than OVO. 

The results could be explained by the high flexibility of the plant protein, which contributes to absorption at the oil-water or air-water interface for a stable interface layer. More high surface hydrophobicity contributes to the stability of foams and emulsions [74], which is correlated with the results obtained for GLO and TE in this study.

## 4. Conclusions

This study applied two extraction methodologies to obtain the total protein fraction (TE) and the individual fractions (ALB, GLO, and GLU) from defatted chickpea flour. It showed that simple and low-cost extraction methods (alkaline solubility and solvent solubility) could obtain a high protein yield with a significant contribution of essential and hydrophobic amino acids. When evaluating their physicochemical characteristics, a potential application of chickpea proteins in food formulations is for both acid and neutral pH, where its high stability was evidenced due to their high Z potential values under these conditions.

In this work, we observed differences in the physicochemical properties (yield, molecular weight, Z potential, and hydrophobic amino acids) of the total protein fraction compared to the individual fraction obtained. It was emphasized by analyzing its surface and structural properties compared to the food industry’s most widely used animal protein. The correlation between the secondary structure of proteins concerning their functional properties (formation of foams and emulsions) was evidenced, which is crucial for the food industry when formulating products based on vegetable proteins in replacement of animal protein with functional features.

Chickpea proteins have improved surface properties compared to OVO, evidencing their potential use as foam and emulsion stabilizers. In addition, surface properties were related to the secondary structure of chickpea proteins and their flexibility. Although the best fraction for replacing OVO was the GLO fraction, the cost, yield, and time to obtain it are high. So, the TE fraction could be a better choice for industrial applications.

The food industry is increasingly interested in protein sources with good functional properties and less allergenicity. Therefore, based on our results, chickpea proteins could be a potential replacement alternative to the most widely used animal protein in the food industry, ovalbumin.

## Figures and Tables

**Figure 1 polymers-15-00110-f001:**
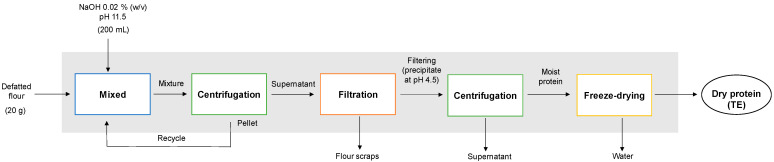
Flow chart describing the total extraction method of chickpea protein.

**Figure 2 polymers-15-00110-f002:**
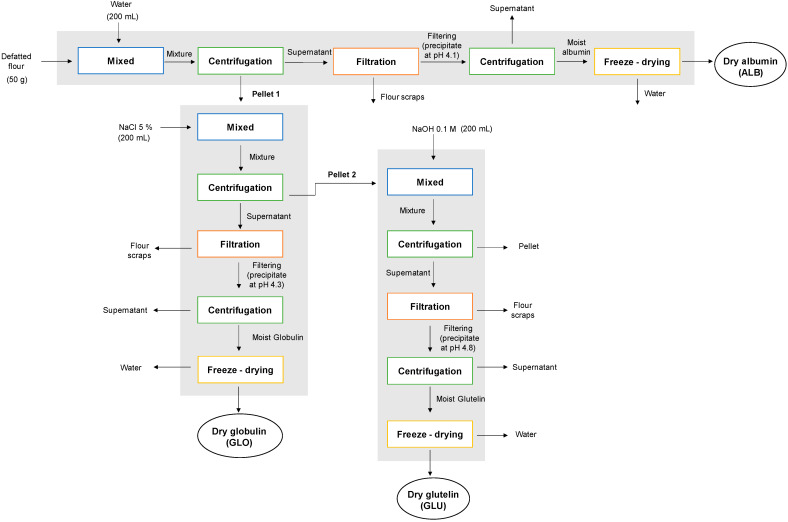
Flow chart describing the sequential extraction method of chickpea protein.

**Figure 3 polymers-15-00110-f003:**
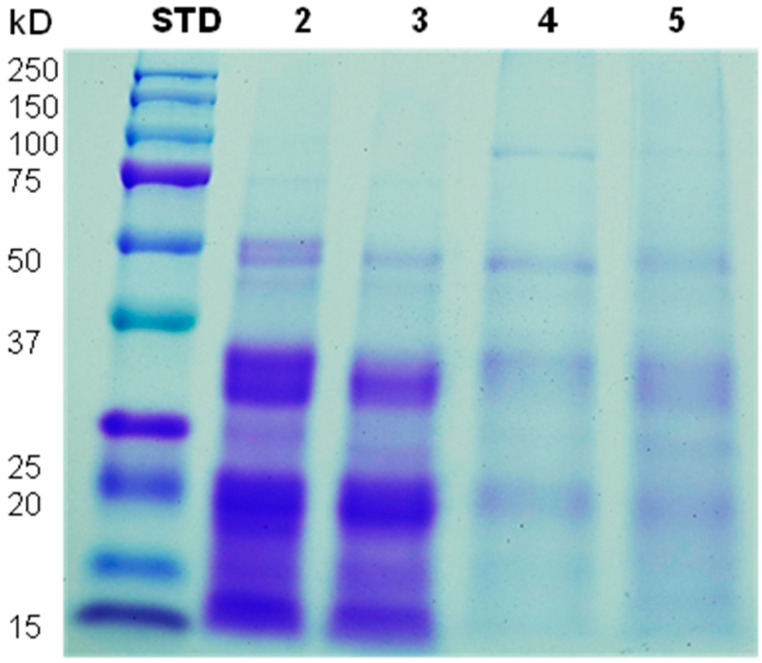
SDS-PAGE of chickpea protein isolates from sequential extractions and ovalbumin. STD: Standard protein markers; (2) Extraction total; (3) Albumin fraction; (4) Globulin fraction; (5) Glutelin fraction.

**Figure 4 polymers-15-00110-f004:**
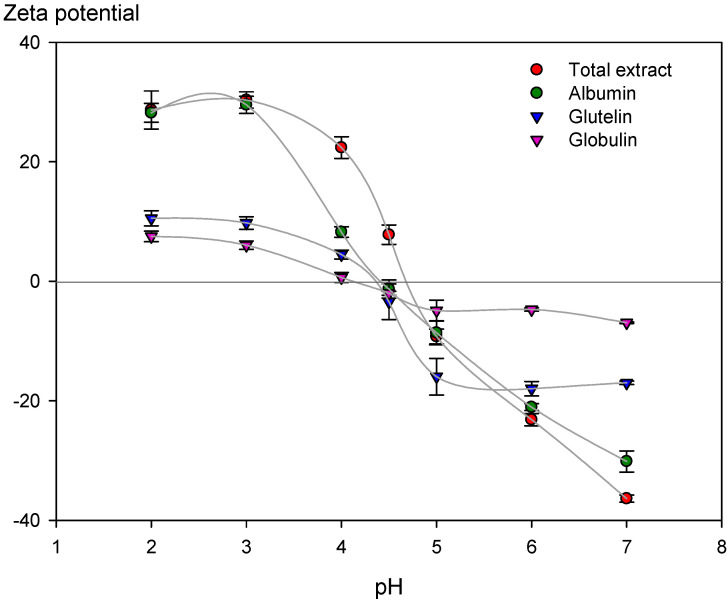
Zeta potential of proteins extracted from chickpea flour. Protein concentration 0.1% (*w/v*).

**Figure 5 polymers-15-00110-f005:**
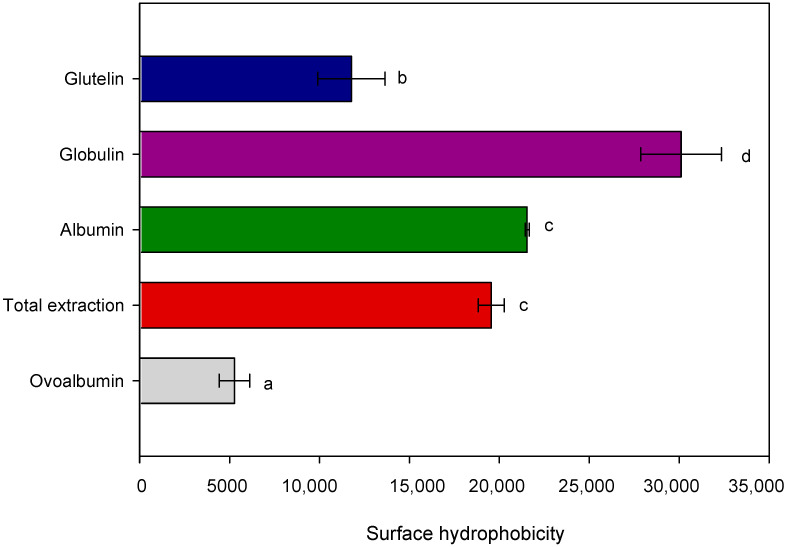
Surface hydrophobicity of proteins extracted from chickpea flour. Different letters (a–d) indicate significant differences (*p* < 0.05) between samples. Error bars indicate the mean and standard deviation of triplicates.

**Figure 6 polymers-15-00110-f006:**
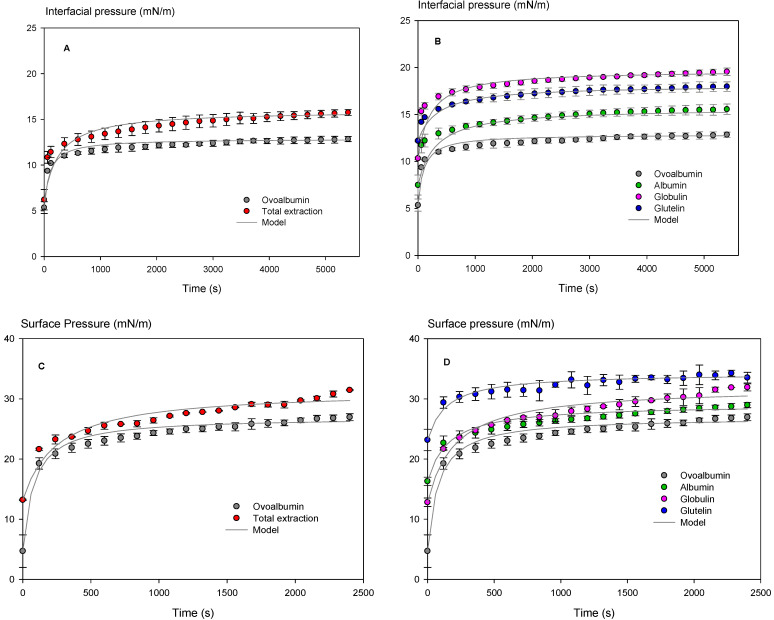
Changes in surface and interfacial pressure between protein dispersions extracted from chickpea flour and commercial ovalbumin. (**A**): Total extraction interfacial pressure concerning ovalbumin; (**B**): Interfacial pressure of albumin, globulin, and glutelin concerning ovalbumin; (**C**): Total extraction surface pressure concerning ovalbumin; (**D**): Surface pressure of albumin, globulin, and glutelin concerning ovalbumin. Protein concentration 0.1% (*w/v*), error bars indicate the mean and standard deviation of triplicates.

**Figure 7 polymers-15-00110-f007:**
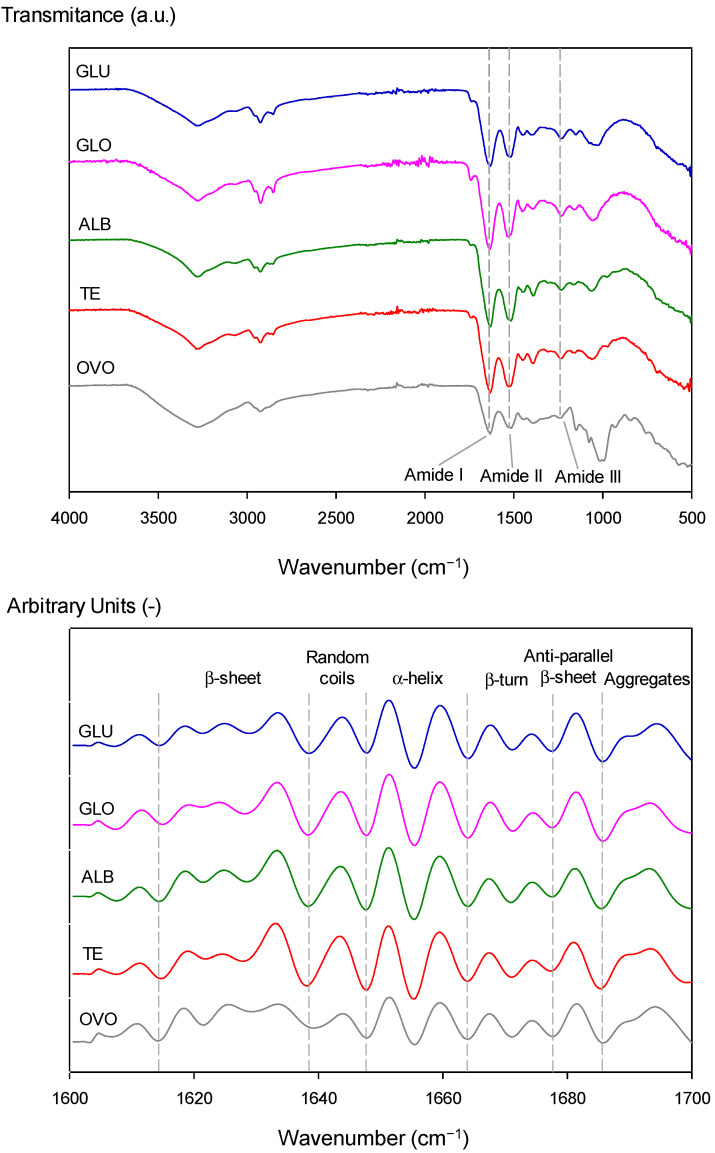
Fourier transform infrared spectra of ovalbumin (OVO), total extraction (TE), albumin (ALB), globulin (GLO), and glutelin (GLU). (**A**) Full spectra and (**B**) Second derivative spectrum.

**Table 1 polymers-15-00110-t001:** Moisture content and a_w_ of lyophilized protein.

Sample	Moisture (%)	a_w_
Total extraction	3.65 ^d^ ± 0.11	0.22 ^c^ ± 0.030
Albumin	1.24 ^a^ ± 0.03	0.05 ^a^ ± 0.004
Globulin	1.91 ^c^ ± 0.44	0.10 ^ab^ ± 0.020
Glutelin	1.87 ^bc^ ± 0.29	0.15 ^b^ ± 0.004

Different letters (^a, b, c, d^) indicate significant differences (*p* < 0.05) between samples.

**Table 2 polymers-15-00110-t002:** Protein extraction yield for total and sequential process extraction.

Sample	Yield Dry Weight (%)
Total extraction	78.0 ^c^ ± 1.0
Total sequential extraction	85.0 ^c^ ± 5.7
Albumin	69.3 ^b^ ± 4.4
Globulin	8.7 ^a^ ± 1.0
Glutelin	7.1 ^a^ ± 0.9

Different letters (^a, b, c^) indicate significant differences (*p* < 0.05) between samples.

**Table 3 polymers-15-00110-t003:** Amino acid profile of chickpea proteins.

Amino Acid	Total Extraction (g/100 g Protein)	Albumin (g/100 g Protein)	Globulin (g/100 g Protein)	Glutelin (g/100 g Protein)
Isoleucine	2.95 ^d^ ± 0.01	2.72 ^c^ ± 0.07	2.23 ^a^ ± 0.01	2.51 ^b^ ± 0.01
Leucine	5.40 ^d^ ± 0.08	4.99 ^c^ ± 0.11	3.82 ^a^ ± 0.07	4.55 ^b^ ± 0.11
Threonine	2.54 ^ab^ ± 0.04	2.44 ^a^ ± 0.08	2.43 ^a^ ± 0.05	2.83 ^b^ ± 0.13
Valine	2.69 ^b^ ± 0.09	2.62 ^b^ ± 0.09	2.15 ^a^ ± 0.08	2.56 ^b^ ± 0.07
Phenylalanine	4.09 ^c^ ± 0.05	3.30 ^b^ ± 0.01	2.65 ^a^ ± 0.00	3.27 ^b^ ± 0.00
Methionine	0.88 ^b^ ± 0.01	0.71 ^a^ ± 0.02	0.70 ^a^ ± 0.01	0.76 ^ab^ ± 0.10
Histidine	1.51 ^b^ ± 0.05	1.53 ^b^ ± 0.08	1.04 ^a^ ± 0.01	1.37 ^b^ ± 0.16
Lysine	5.35 ^b^ ± 0.23	3.83 ^a^ ± 0.11	3.75 ^a^ ± 0.20	4.02 ^a^ ± 0.13
**Total essential amino acids**	25.4 ^c^ ± 0.10	22.14 ^b^ ± 0.69	18.76 ^a^ ± 0.06	21.86 ^b^ ± 0.46
**Essential amino acids (%)**	33.61 ^a^ ± 0.26	32.85 ^a^ ± 1.62	35.91 ^b^ ± 0.05	35.92 ^b^ ± 0.17
Aspartic acid	10.74 ^c^ ± 0.46	10.00 ^c^ ± 0.11	7.01 ^a^ ± 0.04	7.86 ^b^ ± 0.20
Cysteine	1.87 ^b^ ± 0.27	1.27 ^a^ ± 0.00	1.01 ^a^ ± 0.05	1.35 ^a^ ± 0.07
Glutamic acid	14.54 ^c^ ± 0.33	11.51 ^c^ ± 1.30	8.31 ^a^ ± 0.57	9.74 ^b^ ± 0.14
Alanine	3.20 ^a^ ± 0.13	3.15 ^a^ ± 0.10	2.92 ^a^ ± 0.02	3.16 ^a^ ± 0.06
Serine	4.63 ^c^ ± 0.26	4.38 ^bc^ ± 0.18	3.30 ^a^ ± 0.08	3.96 ^b^ ± 0.09
Glycine	3.07 ^a^ ± 0.16	3.11 ^a^ ± 0.18	2.26 ^a^ ± 0.09	3.14 ^a^ ± 0.03
Tyrosine	1.53 ^a^ ± 0.13	1.22 ^a^ ± 0.05	1.29 ^a^ ± 0.05	1.56 ^a^ ± 0.17
Arginine	7.46 ^b^ ± 0.38	6.91 ^b^ ± 0.02	4.42 ^a^ ± 0.67	4.91 ^a^ ± 0.13
Proline	3.14 ^b^ ± 0.29	3.73 ^c^ ± 0.07	2.61 ^a^ ± 0.08	3.30 ^bc^ ± 0.03
**Total non-essential amino acids**	50.17 ^d^ ± 0.38	45.28 ^a^ ± 1.93	33.48 ^b^ ± 0.17	38.99 ^c^ ± 0.53
**Non-essential amino acids (%)**	66.39 ^b^ ± 0.26	67.17 ^b^ ± 1.62	64.09 ^a^ ± 0.05	64.08 ^a^ ± 0.17

Different letters (^a, b, c, d^) indicate significant differences (*p* < 0.05) between samples.

**Table 4 polymers-15-00110-t004:** Comparison of the effect of surface and interfacial pressure of chickpea proteins concerning ovalbumin.

	Interfacial Pressure (mN/m)			Surface Pressure (mN/m)		
Sample	k (s^−1^)	Π_eq_ (mN/m)	RMS (%)	k (s^−1^)	Π_eq_ (mN/m)	RMS (%)
Ovalbumin	0.0099 ^c^ ± 0.0003	12.88 ^a^ ± 0.23	3.37	0.0123 ^d^ ± 0.0010	26.96 ^a^ ± 0.59	3.40
Total extraction	0.0038 ^ab^ ± 0.0006	15.97 ^b^ ± 0.17	5.60	0.0040 ^a^ ± 0.0001	31.46 ^c^ ± 0.10	3.87
Albumin	0.0033 ^a^ ± 0.0004	16.16 ^b^ ± 0.15	5.31	0.0064 ^bc^ ± 0.0009	29.12 ^b^ ± 0.61	2.75
Globulin	0.0046 ^b^ ± 0.0007	19.70 ^c^ ± 0.27	3.89	0.0049 ^ab^ ± 0.0006	32.01 ^c^ ± 0.48	3.32
Glutelin	0.0029 ^a^ ± 0.0002	18.12 ^d^ ± 0.53	1.17	0.0071 ^c^ ± 0.0009	34.29 ^d^ ± 0.42	1.83

Different letters (^a, b, c, d^) indicate significant differences (*p* < 0.05) between samples.

## Data Availability

The data presented in this study are available on request from the corresponding author. The data are not publicly available due to it is part of Daniela Soto-Madrid Ph.D. thesis, which is not published yet.

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
