# Peer review of "Structural and Physicochemical Characterization of Extracted Proteins Fractions from Chickpea (Cicer arietinum L.) as a Potential Food Ingredient to Replace Ovalbumin in Foams and Emulsions"

_polymers, 2022, doi:10.3390/polym15010110_

Round 1

Reviewer 1 Report

The paper presented by Soto-Madrid and co-authors shows results regarding the surface and structural properties of total and sequential protein fractions isolated from defatted chickpea flour. In addition, they compare results from chickpea flour protein with ovalbumin, which is the animal protein commonly used in the food industry. Their results suggest that chickpea proteins are an alternative to ovalbumin and can be used as foam and emulsion stabilizers.

The paper is quite detailed, fairly well-referenced, and relevant to the field. This study is attractive to readers of Polymers, but some major points must be addressed before acceptance (see the file attached).

Author Response

Dear Reviewers:

Authors thank you for your contribution with the comments to improve the manuscript. They are considered on the revised version, which is marked in red together with other minor mistakes. Also, we respond to your comments point by point.

Best regards,

Response to Reviewer 1 Comments

General Reviewer comment: The paper presented by Soto-Madrid and co-authors shows results regarding the surface and structural properties of total and sequential protein fractions isolated from defatted chickpea flour. In addition, they compare results from chickpea flour protein with ovalbumin, which is the animal protein commonly used in the food industry. Their results suggest that chickpea proteins are an alternative to ovalbumin and can be used as foam and emulsion stabilizers.

The paper is well-written, quite detailed, fairly well-referenced, and relevant to the field. This study is attractive to readers of Polymers, but some major points must be addressed before acceptance.

Point 1: On line 55: Replace “seeds” with seed

Response 1: The word "seeds" was replaced with "seed" now on lines 64.

Point 2: On lines 162 and 165: Replace “HCl 1 M” with 1 M HCl.

Response 2: In section 2.2.2, “HCl 1M” was replaced with “1 M HCl”.

Point 3: On line 193: Replace “Sample” with Samples.

Response 3: The “sample” was replaced with “samples” in the SDS-PAGE procedure.

Point 4: On lines 202-203: The sentence “Then, the destain solution I was removed, and ... for 12 h at room temperature.” is confusing. Please rewrite it.

Response 4: The sentence was modified in the text, where some part was removed to avoid confusion (lines 213- 217).

Point 5: On lines 239-240: Replace “(TE, ALB, GLO, GLU)” with (TE, ALB, GLO, and GLU).

Response 5: TE, ALB, GLO, GLU, was replace with “TE, ALB, GLO, and GLU”, now in line 251.

Point 6: On lines 271, 279, 288, and 293: Replace “Where:” with where (in lower case and without the colon)

Response 6: “Where” was replaced with “where” and without the colon in all text that it was required.

Point 7: On line 273: Replace “25 °C and” with 25 °C, and.
Response 7: The comma was added before "and", now in line 289.

Point 8: On line 280: Replace “(mN/m) and” with (mN/m), and

Response 8: The comma was added before "and", now in line 297.

Point 9: On lines 283-284: Remove the space between Π and *. Also, remove the italics.

Response 9: The space and the italics were removed.

Point 10: Please check the asterisk size in Π/Π ratio.

Response 10: it was modified and it was checked this in all text.

Point 11: Please check that the equation is correct. From what has been described by the

authors (“substituting Equation 4 in Equation 3 and rearranging terms ...”), I believe that the   factor must also multiply the Π0. If so, how would the fit in the graphs in Figure 6 be affected?

Response 11: The reviewer is right. There was a mistake in the redaction of the methodology, which lead to a misunderstanding in the interpretation of Equation 5. The empirical Equation 5 considered the conditions employed in the works of Azuara and Moyano but was not developed from the same mathematical approach.

Point 12: On line 293: Replace “value and” with value, and.

Response 12: The comma was added.

Point 13: On line 301: Replace “400 cm at −1 a” with 400 cm−1 at a.

Response 13: It was a mistake and “at” was delated.

Point 14: On line 332: Replace “aw” with aw.
Response 14: Changed.

Point 15: On line 341: Please check the font used in “significant differences (p ≥ 0.05) in the”.

Response 15: the font was corrected to Arial.

Point 16: In subsection 3.1.2, the authors present different SE values from those shown in Table 2. For example, ALB has values of 69.25% and 64% in Table 2 and the text, respectively. The same situation occurs for GLO and GLU fractions. In addition, at the beginning of the subsection, it is reported that the process yield for both extractions was 82±4%, but this value is not shown in Table 2. Was there a typo? Otherwise, I suggest that the authors be clearer in the presentation and discussion of this subsection.

Response 16: Sorry about the confusion. It was clarified in the text. Statistical analysis reported that no significant differences were obtained between TS (78±1%) and total TS (85±6%). For that, an average value was calculated between these values  (82%)  and the standard deviation (±3%) was calculated by partial derivatives. So, we use this average value to discuss our results. To clarify, it was included in subsection 3.1.2

Point 17: In the last paragraph of subsection 3.2.1, it is said that the patterns obtained from SDS-PAGE for the ALB, GLO, and GLU fractions were similar and that the extraction process based on protein solubility was not specific enough. Wouldn’t this statement compromise the other results? The authors could argue a little more about this.

Response 17: The statement was changed argue more about the result obtained from SDS-PAGE, where the other results are not compromise due to physico-chemical properties were different between protein fractions.  So, the similar bands observed were attributed to a low resolution more than not enough specific solubility.

The text was modified and mark in red in the manuscript revised in section 3.2.1.

Point 18: Why did the authors classify cysteine as an essential amino acid in Table 3?

Response 18: In the first version of the manuscript, Cysteine was classified as essential aminoacid considering that it is a semi-essential amino acid, following Ghribi et al. (2015), where authors included Cys in equation of essential aminoacid percentage (the same equation used in this manuscript, and included below). However, following definition of essential amino acid, Cys is not considered. For that, the equation was modified delated Cys and values from Table 3 were changed. This modification not altered the results described previously.

Ghribi, A. M., Gafsi, I. M., Blecker, C., Danthine, S., Attia, H., & Besbes, S. (2015). Effect of drying methods on physicochemical and functional properties of chickpea protein concentrates. Journal of Food Engineering, 165, 179-188.

Point 19: In Table 3: Replace the comma (1,53) with the period (1.53) in the first column of the row referring tyrosine.

Response 19: Thanks for noting the mistake. It was corrected in table 3.

Point 20: In Table 3: How did the authors calculate total amino acid values (essential and non-essential)? Was it the sum of the contribution of each amino acid? If so, why do the total values in Table 3 not correspond to this sum? This is also observed in the text when the authors discuss the quantity of hydrophobic amino acids in GLO and GLU fraction proteins (line 421).

Response 20: We thank the Reviewer for your contribution and clarify the following:

  1. i) The calculation of the percentage of essential amino acids (EAA %) was made based on the following equation:

EAA (%) =

*where cysteine ​​was not considered an essential amino acid, according to what was justified in point 18.

  1. ii) The calculation of the percentage of non-essential amino acids (NEAA %) was made based on the following equation:

NEAA (%)=  

A modification was made in table 3, to improve the understanding of obtaining the EAA (%) and NEAA (%).

On the other hand, the amount of hydrophobic amino acids (HAA) is calculated as the individual sum of each of them, through the following equation:

HAA (g/100 g protein)=

Aftabuddin, M., & Kundu, S. (2007). Hydrophobic, hydrophilic, and charged amino acid networks within protein. Biophysical journal, 93(1), 225-231.

All of these equations are not included in the text, but the respective reference (Aftabuddin & Kundu, 2007) were included. Besides, the aminoacid profile section was improved considerably taking into account the changes included in Table 3.

Point 21: On lines 426-430: The authors say that the composition of amino acids determines the ratio of hydrophilic and hydrophobic groups and, as a consequence, affects electrostatic interactions. I suggest that the authors expand this discussion a little further, mentioning, for example, the presence of charged patches along proteins and/or polyelectrolytes and how these patches affect their stability (see, for example, Phys. Chem. Chem. Phys. 2017,19, 23397-23413; Phys . Chem . Chem . Phys . 2021, 23, 27195; J. Am. Chem. Soc. 2022, 144, 4, 18131825).

Response 21: Thank you so much for your contribution. The suggested comment was incorporated into the discussion in section 3.2.2 and 3.2.3.

Point 22: lines 431-433, 436, 442-443, 446, 630, 633, and Figure 4: Replace “Z potential” or “Z-potential” with Zeta potential.

Response 22: It was replaced by “Zeta potential” in all lines and also in figure 4.

Point 23: On line 439: Replace “pH’s” with pHs.
Response 23: it was modified.

Point 24: On lines 449-451: It is true that, at the isoelectric point (IP), there is no net charge on the protein. However, to say that there is no repulsion between molecules is too strong. Some studies demonstrate that proteins adsorb on charged surfaces at their IP, that is, even when their net charge is neutral, second or higher-order electrostatic factors favor their adsorption/interaction (see Langmuir 2018, 34, 51, 1572715738; Annu. Rev. Food Sci. Technol. 2020, 11, 1, 365387; Phys . Chem . Chem . Phys . 2021, 23, 27195; J. Am. Chem. Soc. 2022, 144, 4, 18131825).

Response 24: Thank you very much for your contribution and we included in the Z potential discussion. The Lunked et al (2022) was included in the reference list.

Point 25: On lines 507-508: Replace “with respect concerning to ovalbumin” with concerning ovalbumin.

Response 25: it was modified.

Point 26: On line 509: Replace “(a, bc, d)” with (a, b, c, d).
Response 26: Thanks for noting the mistake. It was corrected.

Point 27: On lines 516-517: Please check the font used in “the most significant effect (p ≤ 0.05). This result agrees with what was observed in Πeq”.

Response 27: Font was modified and also text was modified to avoid confusion:

“For surface pressure (Table 4), TE, ALB, GLO, and GLU presented lower absorption rate values at the interface compared to OVO, GLU being the protein that showed the highest Πeq value (34.29 ± 0.42 mN/m) (Figure 6d) and could positively affect foam stabilization”.

Point 28: Replace “(∼ 1681 cm−1) and” with “(∼ 1681 cm−1), and”

Response 28:  It was corrected.

Point 29: Replace “(17.71 to 18.49%,” with (17.71 to 18.49%),.

Response 29: Thanks for noting the mistake. It was corrected, added parentheses at the end of the % symbol.

Point 30: On lines 592-594: The sentence “Consequently, as β-sheet increase ... (Harnkarnsujarit et al.,2014)” is confusing. Please rewrite it.

Response 30: It is correct. the sentence was confusing, so it was replace to improve the discussion and the reference was not considered and for that, deleted.

Point 31: On lines 594-595: Replace “random coil, β-turn, into” with random coil and β-turn into.

Response 31: It was modified, line 595.

Point 32: On line 611: Replace “of animal of plant” with of animal and plant.

Response 32: It was replaced, line 615.

Reviewer 2 Report

The MS requires some clarity on certain aspects.

a) Page 9, Line no 346-348, the statements carry no meaning.

b) Based on comercial OVO, the performances are compared. 

How can authors be sure that their extracted product is comparable to these OVO? It is already treated.

c) What is the basis of choosing a neutral pH?

d) More explanations are required in case of Z potential as well as hydrophobicity.

e) Page 19, line 629-630, it requires more explanation.

Author Response

Dear Editor:

Thank you for the revision. All reviewer comments are responded point by point in red, below each comment. Besides, the changes in the manuscript are as red marks following the Reviewer's comments and the other minor mistakes observed.

Dear Reviewers:

Authors thank you for your contribution with the comments to improve the manuscript. They are considered on the revised version, which is marked in red together with other minor mistakes. Also, we respond to your comments point by point.

Best regards,

Response to Reviewer 2 Comments

General Reviewer comment: The MS requires some clarity on certain aspects.

Point 1: Page 9, Line no 346-348, the statements carry no meaning.

Response 1: The lines indicated was not in Page 9. However, the statement was modify to clarify.

Point 2: Based on comercial OVO, the performances are compared.

Point 3: How can authors be sure that their extracted product is comparable to these OVO? It is already treated.

Response 2 and 3. Althoug OVO is a commercial protein, it showed similar characteristic properties comparing to chickpea proteins, such as it is a globular protein, high solubility in aqueous media and its isoelectric point at pH 4.6 (Judge et al., 1996). Besides, the idea is to compare the performance against a commercial protein to demonstrate that the studied proteins are suitable to replace it. Obviously, then it is required to scale for obtain high purity and process yield of the legumin proteins.

Point 4: What is the basis of choosing a neutral pH?

Response 4: Considering that pH protein stability of fractions indicate high stability at neutral and acidic pH (Figure 4), the surface hydrophobicity, DIT and DST was measured at pH 7 for potential applications in foods.

Point 5: More explanations are required in case of Z potential as well as hydrophobicity.

Response 5: More explanations was incorporated in the dis         cussion of these parameters, following also reviewer 1 comments.

Point 6: Page 19, line 629-630, it requires more explanation.

Response 6: The page not coincide with the lines described. However, the discussion in Page 19 was improved.

Round 2

Reviewer 1 Report

All my questions were addressed and duly clarified. The manuscript, therefore, is now in good shape, and I can recommend its publication.

Author Response

Dear review:

Authors thank you for your the comments to the manuscript. 

Best regards, 

Reviewer 2 Report

a) What causes the adjustment of pH level of supertenant to 4.5? (Page 3, Line 137)

b) Same page, " Once it finished the freeze-dried process" what does it mean?

c) "Finally, gels are stored in distilled water until 203 take photography." the newly added lines are insignificant. It must be altered.

d) "was 0.82 ± 0.01 g/100 g of sample (w.b.) for TE and not detected for SE"-must be improved as it appears to be meaningless.

e) Most of the newly added portions indicated in red fonts produces more ambiguity what the authors are projecting. It is advisable to look more carefully into these lines as they carry vital part of information.

f) MS requires English editing.

Author Response

Dear Reviewer 2,

Authors thank you for your contribution with the comments to improve the manuscript. We we respond to your comments point by point.

Best regards,  

Point a: What causes the adjustment of pH level of supertenant to 4.5? (Page 3, Line 137)

Response a: The pH was 4.5 due to the methodology used following the Chang et al. (2012) protocol. However, to clarify, we explained it between brackets. Incorporated in red in line 135-136.

Point b: Same page, " Once it finished the freeze-dried process" what does it mean?

Response b: It means after the freeze-drying process finishes. In line 140, this sentence was changed to “after” freeze-drying process.

Point c: "Finally, gels are stored in distilled water until 203 take photography." the newly added lines are insignificant. It must be altered.

Response c: Maybe, the download manuscript showed a mistake because the final manuscript doesn´t have it.

Point d: "was 0.82 ± 0.01 g/100 g of sample (w.b.) for TE and not detected for SE"-must be improved as it appears to be meaningless.

Response d: Sorry for the confusion, it was clarified in line 316-317. The quantity was unique for the defatted flour (0.82 ± 0.01 g/100 g of sample (w.b.)

Point e: Most of the newly added portions indicated in red fonts produces more ambiguity what the authors are projecting. It is advisable to look more carefully into these lines as they carry vital part of information.

Response e: Thank you so much for your recommendation all red sentences were improved to clarify the information. It was maintained in red on the revised manuscript.

Point f: MS requires English editing.

Response f: English was improved and all changes were evidenced in red.

Round 3

Reviewer 2 Report

The Revised MS is OK.